# Persistent TLR4 Activation Promotes Hepatocellular Carcinoma Growth through Positive Feedback Regulation by LIN28A/Let-7g miRNA

**DOI:** 10.3390/ijms23158419

**Published:** 2022-07-29

**Authors:** I-Ting Chen, An-Chieh Cheng, Yi-Ting Liu, Chieh Yan, Yi-Chen Cheng, Chiung-Fang Chang, Ping-Hui Tseng

**Affiliations:** 1Institute of Biochemistry and Molecular Biology, School of Life Science, National Yang Ming Chiao Tung University, Taipei 112, Taiwan; itchen1025@gmail.com (I.-T.C.); anjiecheng0920@gmail.com (A.-C.C.); chelsea2237@gmail.com (Y.-T.L.); ai86109@hotmail.com (C.Y.); angel780911@hotmail.com (Y.-C.C.); 2General Surgery Division, Department of Medical Research, Far Eastern Memorial Hospital, New Taipei City 220, Taiwan

**Keywords:** hepatocellular carcinoma, Toll-like receptors 4, LIN28A, let-7g

## Abstract

Chronic inflammation caused by liver damage or infection plays an important role in the development and progression of hepatocellular carcinoma (HCC). The activation of Toll-like receptors 4 (TLR4) is involved in HCC tumorigenesis. Moreover, high TLR4 expression in HCC has been linked to poor prognosis. Although the expression of TLR4 in HCC is relatively low compared to hematopoietic cells, it is important to explore the molecular mechanism leading to the elevation of TLR4 in HCC. In this study, we aimed to investigate the positive regulating loop for TLR4 expression in HCC in response to chronic inflammation. Our results confirm that the mRNA expression of TLR4 and proinflammatory cytokines, including interleukin 6 (IL6) and C-C motif chemokine ligand 2 (*CCL2*), positively correlate in human HCC samples. High TLR4 expression in HCC is more susceptible to lipopolysaccharide (LPS); TLR4 activation in HCC provides growth and survival advantages and thus promotes tumorigenesis. It has been shown that the LIN28/let-7 microRNA (miRNA) axis is a downstream effector of the TLR4 signal pathway, and let-7 miRNA is a potential post-transcriptional regulator for TLR4. Thus, we investigated the correlation between TLR4 and LIN28A mRNA and let-7g miRNA in HCC clinical samples and found that the expression of TLR4 was positively correlated with LIN28A and negatively correlated with let-7g miRNA. Moreover, by culturing PLC/PRF5 (PLC5) HCC cells in low-dose LPS-containing medium to mimic chronic inflammation for persistent TLR4 activation, the mRNA and protein levels of TLR4 and LIN28A were elevated, and let-7g miRNA was decreased. Furthermore, the 3’ untranslated region (3’UTR) of TLR4 mRNA was shown to be the target of let-7g miRNA, suggesting that inhibition of let-7g miRNA is able to increase TLR4 mRNA. While parental PLC5 cells have a low susceptibility to LPS-induced cell growth, long-term LPS exposure for PLC5 cells leads to increased proliferation, cytokine expression and stemness properties. In conclusion, our studies demonstrate positive feedback regulation for chronic TLR4 activation in the modulation of TLR4 expression level through the LIN28A/let-7g pathway in HCC and suggest a connection between chronic inflammation and TLR4 expression level in HCC for promoting tumorigenesis.

## 1. Introduction

Hepatocellular carcinoma (HCC) is among the most common cancers worldwide, and major risk factors include chronic liver damage, which is mainly due to chronic infection with hepatitis B virus (HBV) and hepatitis C virus (HCV); alcohol abuse; metabolic syndrome; high calorie intake; diabetes; and obesity [1,2,3]. Under chronic damage and infection, inflammatory response is activated to produce pro-inflammatory cytokines or inflammation-related genes through transcription factors such as nuclear factor-κB (NF-κB) and signal transducer and activator of transcription 3 (STAT3). Additionally, several signaling pathways activated by inflammation have been indicated as being involved in carcinogenesis, such as the mitogen-activated protein kinases (MAPKs), NF-κB and phosphoinositide 3-kinase (PI3K)/protein kinase B (Akt) pathways [4].

Toll-like receptors (TLRs) are primary sensors of exogenous pathogen-associated molecular patterns (PAMPs) and endogenous damage-associated molecular patterns (DAMPs), and thereby play important roles in inflammatory responses [5]. As the first member of the TLR family to be discovered, TLR4 has been indicated to be involved in mediating innate immunity and inflammation and is linked to tumorigenesis. In general, TLR4 signaling is mediated by the recruitment of adaptor proteins, such as MyD88 and TRIF, to TIR domains in the cytoplasmic portion of the receptors. The dimerized adaptors, in turn, recruit a variety of signaling proteins that activate different effector pathways, including IκB kinase (IKK), p38 MAPK, c-Jun N-terminal kinase (JNK), PI3K and IKK-like kinases, such as TANK-binding kinase 1 (TBK1) and IKK [6]. In cancer cells such as lung, liver, gastric, pancreatic, ovarian, and colon cancer, TLR4 expression is elevated [7,8,9,10,11]. The activation of TLR4 enhances tumor growth or metastasis and protects cancer cells from apoptosis or immune cell-induced lysis [12,13].

MicroRNAs (miRNAs) are able to regulate the 3′ untranslated regions (3’UTRs) of target mRNA through post-transcriptional modification and participate in numerous physiological and pathological processes. In recent years, miRNAs have been identified as important regulators for TLR pathways [14]. For instance, miR-105 inhibits TLR2 expression in human oral keratinocytes [15], and miR-203 in pancreatic cancer-secreted exosomes inhibits TLR4 in dendritic cells [16]. As the largest and the most studied of all miRNA families, the let-7 family has been indicated as being involved in the regulation of TLR4 expression. In cultured human cholangiocytes, let-7i regulates TLR4 expression via post-transcriptional suppression [17]. let-7e is positively regulated by Akt1 in LPS-treated macrophages and targets TLR4 [18]. In human gastric epithelial cells with *Helicobacter pylori* infection, the TLR4 signaling pathway is activated and promotes LIN28 expression to inhibit let-7b, which is complementary to the 3′UTR of *TLR4* mRNA and regulates TLR4 expression [19].

LIN28, which has two homologs, LIN28A and LIN28B, is an RNA-binding protein and mediates biological functions by binding to the target mRNA or miRNA. LIN28 has been shown to bind to the loop portion of pri-let-7 and pre-let-7 to inhibit the activities of Drosha and Dicer, and mediate the terminal uridylation of pre-let-7, leading to the blockage of matured let-7 [20,21,22,23,24,25]. As the main downstream effector of TLR4-mediated IKK signal, NF-κB directly activates the transcription of LIN28 [26,27]. Meanwhile, it has also been shown that interleukin 6 (IL6), an inflammatory cytokine induced by TLR4, mediates STAT3 activation to promote the transcription of LIN28 [28].

TLR4 is a critical receptor responsible for inflammatory response and is associated with HCC progression [29,30]. The expression of TLR4 has previously been associated with enhanced invasion and metastasis abilities and poor prognosis in HCC [31,32]. Although the expression of TLR4 in HCC cells is relatively low compared to expression levels in hematopoietic cells, we are interested in investigating the molecular mechanism that leads to the elevation in TLR4 expression in HCC under chronic inflammation. As human HCC cells are extremely heterogeneous, the correlation of TLR4 with inflammatory cytokines was investigated; we confirmed that the expression of TLR4 is positively correlated with inflammatory cytokines such as IL6 and C-C motif chemokine ligand 2 (*CCL2*), and that TLR4 activation promotes HCC. Next, to explore the post-transcriptional regulation of TLR4 expression, the LIN28/let7 axis, which is involved in tumor development and is associated with inflammation, is a potential molecular target. Thus, the relationship between TLR4 and LIN28/let-7 was investigated. In HCC tissues, a positive correlation was found between *TLR4* and *LIN28A*, while a negative correlation was found between *TLR4* with *let-7g*. With long-term TLR4 activation in PLC5 cells, LIN28A is increased and *let-7g* miRNA is decreased. Moreover, we identified that let-7g directly targets *TLR4* mRNA and inhibits TLR4 expression. Our results suggest that long-term inflammation leads to the upregulation of TLR4 expression and downstream effectors through a positive feedback loop and contributes to HCC development.

## 2. Results

### 2.1. The Expression of TLR4 Is Positively Correlated with IL6 and CCL2 in Clinical Samples

First, the correlation between *TLR4* and *IL6* or *CCL2* levels was investigated. Using the 20 HCC tissues that were collected from HBV- and HCV-negative male HCC patients by the Taiwan Liver Cancer Network, the mRNA expression of *TLR4*, *IL6* and *CCL2* was determined using the quantitative polymerase chain reaction (qPCR). The results show that the expression levels of *TLR4*, *IL6* and *CCL2* varied between human HCC samples, indicating that the upregulation of TLR4, IL6 and *CCL2* might not be a typical event in the early stages of HCC (stage I or II) (Appendix A). We then used Pearson’s correlation analysis to test whether there was a significant relationship between the mRNA expression levels of *TLR4*, *IL6* and *CCL2*. Our results show a positive correlation between the mRNA expression levels of *TLR4* and *IL6* (R = 0.8419, *p* < 0.0001) and *CCL2* (R = 0.8274, *p* < 0.0001) (Figure 1).

### 2.2. Activation of TLR4 in HCC Leads to Survival Advantage

Previous studies have shown that TLR4 is critical for promoting HCC tumorigenesis [26,28]. By screening the expression levels of *TLR4* mRNA and protein in five different liver cancer cells, Huh7, PLC/PRF5 (PLC5), Hep3B, HepG2 and SK-Hep1, HepG2 and Hep3B were examined. Huh7, PLC5, Hep3B and HepG2 cell lines were derived from biopsies of HCC patients, and SK-Hep1 cells were derived from the ascitic fluid of a liver adenocarcinoma patient. Five HCC cell lines with different genetic backgrounds, such as p53 expression and HBV infection, exhibited different levels of TLR4 expression. Huh7 and Hep3B cells expressed high levels of *TLR4* mRNA, while PLC5, HepG2 and SK-Hep1 cells had relatively low *TLR4* mRNA expression. For protein levels, Huh7 cells expressed high levels of TLR4, while PLC5, Hep3B, HepG2 and SK-Hep1 cells demonstrated low TLR4 expression levels (Appendix A). Hep3B cells show inconsistent expression levels between mRNA and protein; thus, Huh7 cells were used to represent cells with high TLR4 expression levels, while PLC5 cells were used to represent cells with low TLR4 expression levels.

By activating TLR4 signaling with lipopolysaccharide (LPS) treatment, Huh7 cells demonstrated approximately 20 percent higher cell viability and a 25 percent increase in colony formation; however, no significant difference in cell viability or colony formation was found in PLC5 cells (Figure 2A,B). Meanwhile, Huh7 cell growth promoted by LPS was diminished in TLR4-knockdown cells (Appendix A), indicating an LPS-mediated enhancement in cell viability and TLR4-dependent colony formation in HCC cells. The downstream signaling pathways of TLR4, such as NF-κB, MAPKs and PI3K pathways, can further help cell growth and proliferation, and may also assist in the progression of cancer. We confirmed that TLR4 activation by LPS in Huh7 cells with high TLR4 expression, not PLC5 cells, was able to mediate the production of inflammatory cytokines and chemokines, including tumor necrosis factor (TNF), IL6, and *CCL2* and Cyclooxygenase 2 (COX2) (Figure 2C), and the activation of Akt, JNK, IKK (Figure 2D,E) and STAT3 signals (Figure 2F), but this was not the case in PLC5 cells with low TLR4 expression. Moreover, LPS treatment is able to promote tumor growth in vivo in a Huh7 xenograft mice model (Figure 2G).

### 2.3. Analysis of the Correlation between TLR4 and LIN28A/Let-7g Axis in HCC Clinical Samples

It has previously been demonstrated that the LIN28 family is transcriptionally regulated via direct promoter binding by transcription factors, such as NF-κB and STAT3, the downstream effectors of TLR4 signaling [26,27,28]. We confirmed that TLR4 activation by LPS in Huh7 cells leads to increases in *LIN28A* mRNA and that LPS-mediated induction of *LIN28A* expression is diminished in TLR4-knockdown cells (Figure 3A). To examine whether NF-κB and STAT3 were responses to the induction of *LIN28A* mRNA, Huh7 cells were treated with LPS in the presence of various inhibitors to block the IKK, PI3K, MAPK and STAT3 pathways. After 4 hours of treatment with dimethyl sulfoxide (DMSO) or inhibitors, no significant effect on cell viability was found (Figure 3B). The results show that the induction of *LIN28A* mRNA was blocked by an IKK inhibitor, BMS345541, and a STAT3 inhibitor, WP1066, indicating that LPS-mediated LIN28A expression is NF-κB- and STAT3-dependent (Figure 3C). Previous studies suggested that Lin28A inhibits biogenesis of *let-7* miRNA by the direct binding to pre-let-7 or pri-let-7 and inhibits its processing [20,21,22,23,24,25]. Therefore, the inhibition of IKK or STAT3 pathways with BMS345541 or WP1066 leads to the elevation of *let-7g* miRNA (Figure 3D).

To clinically investigate whether TLR4 mRNA expression is associated with the expression *LIN28* mRNA or *let-7* miRNA, 20 human HCC tissues (Figure 1) were further analyzed for the expression of *LIN28A* and *LIN28B* mRNA and *let-7b*, *let-7i* and *let-7g* miRNA (Appendix A). We focused on LIN28A for the correlation analysis due to the low expression level of *LIN28B* in clinical samples. By employing Pearson’s correlation analysis, we found that the expression level of *LIN28A* mRNA was also inversely related to *let-7g* miRNA (R = −0.7259, *p* < 0.001; Figure 4A), but not *let-7b* and *let-7i* miRNA (R = −0.1402 and −0.1213, respectively). By analyzing *TLR4*’s association with *LIN28A* mRNA and *let-7g* miRNA, we found that there was a positive correlation between *TLR4* and *LIN28A* mRNA level (R = 0.7975, *p* < 0.0001), and an inverse correlation between TLR4 mRNA and *let-7g* miRNA (R = −0.6662, *p* < 0.005; Figure 4B,C). In addition to human HCC samples, tumor cells from a model with DEN-induced HCC mice were isolated and analyzed for expression of *Tlr4* and *Lin28a* mRNA and *let-7g* miRNA level (Appendix A). The results show that Tlr4 mRNA level was positively correlated with *Lin28a* (R = 0.8972, *p* < 0.001), and inversely correlated with *let-7g* miRNA (R = −0.7836, *p* < 0.01; Appendix A). Based on these results, we demonstrated that *TLR4* and *LIN28A* mRNA are positively correlated, while *TLR4* and *let-7g* miRNA are negatively correlated, confirming the association between TLR4 and the LIN28A/let-7g axis.

### 2.4. HCC with Long-Term TLR4 Activation Leads to Upregulation of TLR4 and LIN28A and Downregulation of Let-7g miRNA

It has also been shown that *let-7* miRNA, which is inhibited by *LIN28*, might target TLR4 mRNA. Based on the correlation between *TLR4*, *LIN28A* and *let-7g* clinically, we investigated whether a positive feedback loop links chronic inflammatory response with upregulation of TLR4 expression in HCC cells. Human PLC5 cancer cells with low TLR4 expression were cultured in the presence of LPS (10 ng/mL) to mimic chronic inflammatory response. After 10 and 20 generations of cultivation, the results show that the expression of *TLR4* and *LIN28A* mRNA was increased in PLC5 cells cultured with LPS-containing medium (Figure 5A,B). The protein expression levels of TLR4 and LIN28A were detected by Western blotting (Figure 5C). The results show that TLR4 and LIN28A expression levels were increased with long-term LPS exposure (Figure 5D). Next, the expression level of *let-7g* miRNA was examined. Our results show that PLC5 cells under chronic inflammation increased the expression of TLR4 and LIN28A and caused a reduction in *let-7g* miRNA (Figure 5E). Additionally, *c-MYC*, which is an important transcription factor for stemness and is inhibited by *let-7* miRNA, was analyzed to confirm the downregulation of let-7 miRNA. The data show that *c-MYC* increased in PLC5 cells after long-term LPS exposure (Figure 5F). As the expression of TLR4 gradually increased under the influence of long-term LPS treatment, the activation of TLR4-induced signaling was examined. The experimental results show that, compared with PLC5 cells that had not undergone long-term LPS exposure, TLR4-mediated signaling, including Akt and p38, in PLC5 cells with long-term LPS exposure was activated in response to LPS stimulation for 30 min (Figure 5G), indicating that long-term LPS stimulation to mimic the chronic inflammatory environment is able to promote TLR4 expression and sensitize TLR4-induced signals in HCC cells.

### 2.5. TLR4 Is the Direct Target to Be Regulated by MicroRNA Let-7g

In epithelial cells, it was found that *let-7i* or *let-7b* miRNAs target *TLR4* mRNA and regulate the expression of TLR4 [17,19]. With the high sequence similarity of let-7g with let-7b and let-7i, it is likely that *let-7g* miRNA can regulate the expression of *TLR4* mRNA. With microRNA.org (http://www.microrna.org) (accessed on 20 September 2018), the 3′UTR of TLR4 mRNA was predicted as a potential seed region for *let-7g* miRNA (Figure 6A). To determine the relationship between *let-7g* miRNA and *TLR4* mRNA, let-7g was transfected into Huh7 cells, and the expression level of *TLR4* mRNA and protein level were examined. It can be seen that *TLR4* mRNA was inhibited by *let-7g* miRNA (Figure 6B). Meanwhile, the protein expression level of TLR4 decreased with the overexpression of *let-7g* miRNA (Figure 6C,D). To further verify that let-7g regulates the 3′UTR of TLR4 mRNA through post-transcriptional modification, a luciferase reporter assay was performed. It was found that luciferase activity in the construct with the 3′UTR of *TLR4* mRNA was reduced by about 50% compared with the empty construct without the 3′UTR of *TLR4* mRNA in the presence of pre-*let-7g* miRNA (Figure 6E). With the mutated 3′UTR of *TLR4* in the binding site of let-7g, luciferase activity was not decreased by pre-let-7g miRNA (Figure 6E). These data indicate that let-7g inhibits *TLR4* mRNA through post-transcriptional modification.

### 2.6. HCC with Long-Term LPS Exposure Leads to Increase Proliferation and Stemness Properties

Based on our results, it was observed that under the influence of chronic inflammation, the expression of TLR4 gradually increased. Therefore, we investigated whether the increase in TLR4 expression promoted HCC tumorigenesis. Compared with control PLC5 cells, PLC5 cells with long-term LPS exposure exhibited higher cell viability in the MTT assay and promoted colony formation ability (Figure 7A,B). Furthermore, TLR4 activation leads to the downregulation of *let-7* miRNA, which is responsible for controlling stemness properties. Elevation of TLR4 expression in LPS-treated PLC5 cells promotes the formation of spheroid cells, indicating an increase in cancer precursor cells (Figure 7C). The spheres were collected and the mRNA levels of stemness-associated genes, including *NANOG* and *OCT4*, were analyzed. Consistent with the sphere formation assay, *NANOG* and *OCT4* mRNA levels were increased in the spheres of LPS-exposed PLC5 cells (Figure 7D). Next, tumor growth in the xenograft mice model was examined. From the results, it can be seen that tumor growth increased with PLC5 cells cultured in LPS-containing medium for 20 passages compared with tumors with PLC5 cells cultured in control medium for 20 passages, suggesting that HCC cancer cells cultured under chronic inflammation promote tumorigenesis (Figure 7E).

## 3. Discussion

The inflammatory response is an important defense mechanism that impacts every single step of tumorigenesis [4]. It is known that chronic inflammatory response is a risk factor for liver cancer progression. TLR4 plays important roles in innate and inflammatory responses and is a crucial receptor to sense PAMPs and DAMPs. As the first line of defense against gut-derived antigens, emerging evidence has shown that activation of TLR4 signaling in liver contributed to inflammation- and injury-induced HCC promotion [33]. Based on our results from human HCC samples, the expression of *TLR4* is positively correlated with *IL-6* or *CCL2* level, indicating that the expression of TLR4 is associated with inflammatory effectors. In line with published results, we showed that the activation of TLR4 signaling in HCC provides growth advantages by promoting cytokine expression and proliferation through the Akt and STAT3 pathways.

Signaling pathways that mediate the protumorigenic effects of inflammation are often subject to a feed-forward loop [2]. A recent study has shown that TLR4 activation promotes HCC progression through a COX-2/prostaglandin E2 (PGE2)/STAT3 positive feedback loop [34]. This study focused on investigating the mechanism that regulates TLR4 expression level in HCC in response to chronic inflammation. With our results, the activation of TLR4 triggers NF-κB-mediated inflammatory response, while LIN28 and *let-7* miRNA are indicated in tumor development and are associated with inflammation. A positive correlation was found between *TLR4* and *LIN28A*, while a negative correlation was demonstrated between *let-7g* miRNA and *TLR4* or *LIN28A*. To mimic chronic inflammation with persistent TLR4 activation, we cultured PLC5 cells in low-dose LPS-containing medium; in this way, we demonstrated that mRNA and protein levels of TLR4 and LIN28A were elevated. The increase in LIN28A led to a decrease in *let-7g* miRNA. Furthermore, the 3′UTR of *TLR4* mRNA was shown to be the target of *let-7g* miRNA, suggesting that the inhibition of let-7g miRNA was able to increase *TLR4* mRNA.

The LIN28/let-7 axis has previously been highlighted in cancer progression [35]. It can be found in many cancers (e.g., breast cancer, lung cancer, or prostate cancer) that LIN28-positive tumors are more aggressive, and the inhibition of *let-7* miRNAs promotes the development of cancer [24]. The LIN28/let-7 molecular switch has emerged as a central regulator of liver diseases, including liver injury by viral infection, hepatitis, cirrhosis, and HCC [25]. Increased LIN28 and decreased let-7 expression in HCC are pathologically associated with poor prognosis [36,37]. At present, 13 members of the let-7 family have been identified in humans, including let-7a-1, let-7a-2, let-7a-3, let-7b, let-7c, let-7d, let-7e, let-7f-1, let-7f-2, let-7g, let-7i, miR-98, and miR-202 [38]. Oncogenic signaling pathways, such as c-Myc, Ras, high-mobility group A (HMGA), and signal transducer and activator of transcription 3 (STAT3), which are critical in tumorigenesis, proliferation and invasion, are targeted by let-7 and, thus, let-7 is associated with stem cell differentiation and tumor suppression [39,40,41,42].

In this study, we demonstrated that HCC with long-term LPS exposure leads to increased cell proliferation and stemness. It has been reported previously that LPS could also affect tumor immunity. In addition, it has been observed that LPS leads to the differentiation of hepatic progenitors into myofibroblasts and increases the production of IL6 and TNF [43]. LPS is a trigger for macrophage-mediated inflammation, and LPS-mediated TLR4 activation induces pro-inflammatory cytokines and interferon (IFN)/IFN inducible genes [44,45].

In addition to the chronic inflammatory microenvironment, the genes of interest in this study, including *TLR4*, *LIN28A* and *let-7g* miRNA, have been demonstrated to be involved in HCC tumorigenesis in several studies in the literature. However, this study elucidates the potential molecular mechanisms that modulate TLR4 expression in HCC through a positive feedback loop by the TLR4 and LIN28A/let-7g pathways in response to persistent TLR4 activation. Importantly, HCC cells with elevated TLR4 expression are sensitive to TLR4-mediated tumor promotion through the activation of Akt, NF-κB and STAT3 signaling and the downregulation of *let-7* miRNA (Figure 8). These findings help to shape new concepts for the regulation of TLR4 expression in HCC through chronic inflammation and provide new insights for HCC clinical treatment. Furthermore, the genotype and phenotype of human HCC are variable and heterogenous; thus, it would be of interest to further investigate the clinicopathological features of HCC patients with elevated TLR4 in the future.

## 4. Materials and Methods

### 4.1. Reagents and Antibodies

The following antibodies were used: anti-TLR4 (sc-10741), anti-JNK (sc-7345), anti-IκBα (sc-371), anti-α-tubulin (sc-23948) and anti-α-actin (sc-69879) from Santa Cruz (Dallas, TX, USA); HRP-conjugated anti-rabbit light chain (MAB201P) and HRP-conjugated anti-mouse light chain (AP200P) from EMD Millipore (Burlington, MA, USA); and anti-Akt (#9272), anti-LIN28A (#3978), anti-phospho-Akt (Thr308) (#2965), and anti-phospho-SAPK/JNK (Thr183/Tyr185) (#9251) from Cell Signaling (Danvers, MA, USA).

### 4.2. Study Subjects

The tumor tissues were collected from Stage I or II (based on The American Joint Committee on Cancer (AJCC) TNM system), HBV- and HCV-negative male patients that had not received chemotherapy or radiotherapy in the Taiwan Liver Cancer Network (TLCN). The HCC samples were used to clinically examine the expression of *TLR4*, *IL6*, *CCL2*, *LIN28A* and *LIN28B* mRNA and *let-7b*, *let-7g* and *let-7i* miRNA.

### 4.3. Cell Culture and Treatment

Human HEK293T and human PLC/PRF5 (PLC5) cells were obtained from the Bioresource Collection and Research Center (BCRC, Hsinchu, Taiwan). Human Huh7 cells were obtained from the Japanese Collection of Research Bioresources (JCRB, Tokyo, Japan). The cells were grown in Dulbecco’s modified Eagle medium (DMEM) containing 10% fetal bovine serum (FBS), 1 mM sodium pyruvate, and 1% (*v*/*v*) penicillin-streptomycin solution. DMEM, sodium pyruvate and penicillin–streptomycin solution were obtained from Biological Industries (Kibbutz Beit Haemek, Israel), and FBS was purchased from Invitrogen (Waltham, MA, USA).

### 4.4. Plasmids

The lentivirus-based pLKO.1-shTLR4 (TRCN0000056894 and TRCN0000056897) for human TLR4 shRNA-expression, pLKO.1-shLuc (TRCN0000072243) for luciferase shRNA-expression, pLKO-AS1010 for expressing indicated microRNA, and the packaging plasmids, pCMVΔ8.91 and pMD.G, were obtained from the National RNAi Core Facility (Institute of Molecular Biology/Genomics Research Center, Academia Sinica, Taipei, Taiwan). pRL-TK-Renilla luciferase and pGL3-control vector, used for the luciferase reporter assay, were from Promega (Madison, WI, USA). All constructs were confirmed by DNA sequencing.

### 4.5. Real-Time Quantitative Polymerase Chain Reaction

RNA samples were isolated with TRIzol (Invitrogen), and reverse transcribed to synthesize cDNA with 100 nM of each of the RT oligonucleotides (for RT of miRNA) or 10 µM of random hexamer (for RT of mRNA) by the iScript cDNA or iScriptTM select cDNA synthesis kit (Bio-Rad, Hercules, CA, USA). Target mRNA was quantified by real-time quantitative polymerase chain reaction (RT-qPCR) with the StepOnePlus system (Applied Biosystems, Waltham, MA, USA). Primer sequences are available in Appendix A. All values were normalized to the level of human GAPDH or mouse Cph mRNA expression.

### 4.6. Cell Viability Assay

The cell viability was determined using the MTT assay. Cells were seeded onto a 96-well plate at 5 × 10^3^ cells per well. Twenty-four hours later, the culture medium was replaced with LPS for the indicated concentration. After incubating for 4 days, 30 μL of MTT solution (4 mg MTT/mL PBS) was added into each well and incubated for 4 h. The medium was removed, and Formazan crystals formed by the cells were dissolved using 200 μL of DMSO. The absorbance was read at 560 nm using 630 nm as the reference wavelength on a Multiwell plate reader.

### 4.7. Colony Formation Assay

Huh7 or PLC5 cells were seeded onto a 6-well plate at 1 × 10^3^ cells per well. After incubating for two weeks, the colonies formed were fixed in 3.7% formaldehyde solution and visualized by staining with 0.2% crystal violet. The number of colonies was counted under the microscope. Data were obtained from 3 independent experiments.4.8. Lentivirus Production and Infection

HEK293T cells were transfected with the lentivirus-based constructs along with the packaging plasmids, pCMVΔ 8.91 and pMD.G, using T-pro NTRII (T-Pro Biotechnology, New Taipei City Taiwan). Virus-containing medium was collected at 48 and 72 h post-transfection. Cells were infected with lentivirus-containing medium at a multiplicity of infection (MOI) of 10 to 25 in the presence of 10 μg/mL polybrene (Sigma-Aldrich, St. Louis, MO, USA). After 24 h, the virus-containing medium was replaced with selection medium containing 5 μg/mL puromycin (EMD Millipore). After cell growth was stable, cells were used in subsequent experiments.

### 4.8. Mouse Mammary Tumor Model and Xenograft Tumor Formation

For the DEN-induced HCC model, three-week-old male C57/BL6 mice were injected intraperitoneally (i.p.) with 5 mg/kg of DEN (Sigma-Aldrich). After 8 months on normal chow, mice were sacrificed, and HCC cells were isolated as described by He et al. [46]. For the xenograft tumor model, six-week-old male SCID mice were obtained from the National Laboratory Animal Center (Taipei, Taiwan). Each mouse was inoculated s.c. in the right dorsal flank with indicated cancer cells suspended in 50 μL PBS and an equal volume of Matrigel (BD Biosciences, San Jose, CA, USA). Tumors were measured with Vernier calipers. Tumor volumes were calculated with the following formula: width^2^ × length × 0.52. Whenever tumor size reached 2000 mm^3^ before the end of treatment, mice were sacrificed based on the Animal Care and Use Guidelines.

### 4.9. Immunoblotting

Cells were lysed in ice-cold RIPA lysis buffer containing 50 mM Tris-HCl, pH 8.0, 150 mM NaCl, 1% NP40, 0.5% deoxycholate, 0.1% SDS, 1 mM PMSF, 1 mM Na_3_VO_4_ and 1 mM NaF. After centrifugation, the proteins in each cell extract were separated by SDS-PAGE, electro-transferred to polyvinylidene difluoride (PVDF) membrane and analyzed by immunoblotting. The membranes were stained with ponceau S to confirm the efficiency and consistency of the protein, blocked with 5% nonfat dry milk in TBST buffer, and then probed with the indicated antibodies. The blots were visualized using a chemiluminescence reagent (Thermo, Rockford, IL, USA) and exposure of an X-ray film (Fujifilm, Taipei, Taiwan). Quantification was carried out with Image J 1.53.

### 4.10. Regrowth Protocol for PLC5 Cells with or without LPS Exposure

PLC5 cells were seeded in a 6 cm dish at a density of 1 × 10^6^ cells per 5 mL of culture medium with or without 10 ng/mL LPS. After 3 days, the cells were trypsinized, counted and subcultured with the indicated density for indicated passages.

### 4.11. Sphere-Formation Assay

PLC5 cells (1 × 10^5^) were seeded in a 10 cm dish and cultured in tumor sphere medium consisting of serum-free DMEM/F12 medium (GIBCO, Waltham, MA, USA), N-2 supplement (GIBCO), 10 ng/mL human recombinant bFGF-basic (PeproTech, Rehovot, Israel), and 10 ng/mL EGF (PeproTech). The medium was changed every other day until tumor sphere formation was achieved in approximately 3 weeks [47].

### 4.12. Luciferase Reporter Assay

As previously described [48], HEK293T cells were seeded onto a 96-well plate at 1 × 10^4^ cells per well, and co-transfected with 10 ng of luciferase reporter and 50 ng of miRNA-expressing construct using T-pro NTRII. After transfection for 24 h, luciferase activity was monitored using the Dual-Glo^®^ luciferase assay system and a luminometer. Renilla luciferase activity was normalized against Firefly luciferase activities and presented as the percentage of inhibition.

### 4.13. Statistical Analysis

Correlation analysis was completed using the Pearson method with SPSS statistical software (IBM SPSS Statistics 20.0, IBM, Armonk, NY, USA). All data were reported as mean ± SD. The comparison of means was performed with multiple *t*-tests using GraphPad Prism version 6.01 (GraphPad Software, San Diego, CA, USA). The tumor volume data satisfied the assumptions of normality and homogeneity required for parametric analysis and, thus, group means at the indicated days were compared with a one-way analysis of variance followed by Fisher’s least significant difference method for multiple comparisons using GraphPad Prism (version 6.01). A *p* value of less than 0.05 (* *p* < 0.05, ** *p* < 0.01) was considered statistically significant.

## Figures and Tables

**Figure 1 ijms-23-08419-f001:**
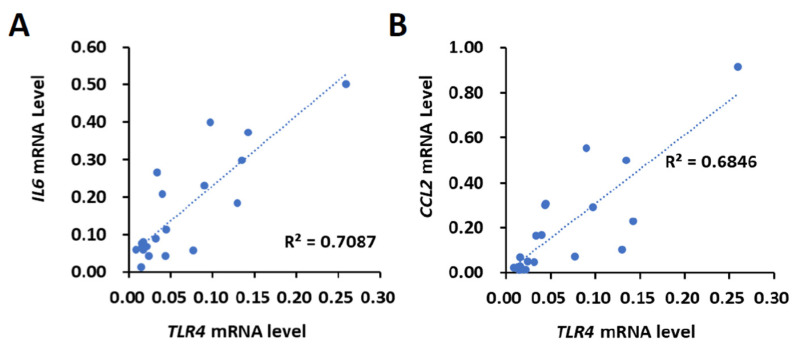
Positive correlation of *TLR4* with *IL6* or *CCL2* mRNA in HCC clinical samples. Correlation between the mRNA expression of *TLR4* with (**A**) *IL6* and (**B**) *CCL2* in HCC patients. Pearson’s correlation analyses were performed. Data are presented as each value (R^2^, correlation coefficient).

**Figure 2 ijms-23-08419-f002:**
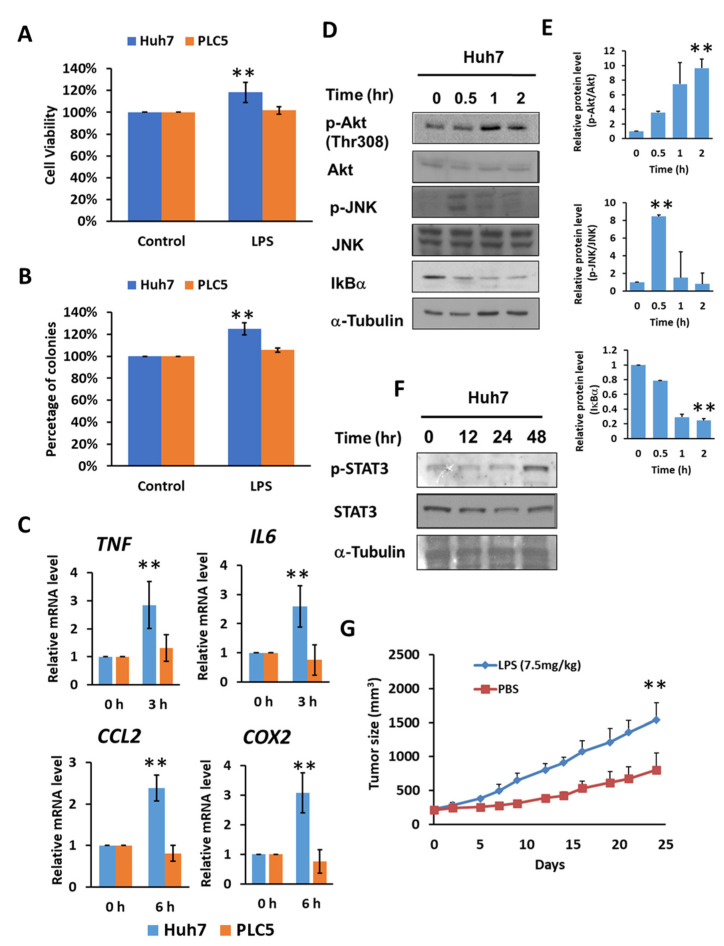
TLR4 activation in Huh7 HCC cells leads to growth and survival advantages. (**A**) Cell viability and (**B**) clonogenic ability of Huh7 and PLC5 cells with TLR4 engagement. Huh7 and PLC5 cells incubated with or without LPS (10 ng/mL) were placed on 96-well plates for 4 days or on 6-well plates for 14 days. Cell viability was measured with MTT assays. Clonogenic ability was determined by colony formation, which was visualized by crystal violet. The relative viability and clonogenic ability of cells without LPS treatment were set as 100 percent. Results are mean ± SD for six separate experiments. (**C**) Induction of *TNF*, *IL6*, *CCL2* and *COX2* mRNA in Huh7 and PLC5 cells with LPS stimulation. Huh7 or PLC5 cells were treated with or without LPS for indicated time periods. RNA was extracted, and the expression of the indicated mRNAs was measured by qPCR. The relative mRNA level in Huh7 cells with LPS treatment at 0 h was set as 1.0. Results are mean ± SD for three separate experiments. (**D**) Activation of Akt, JNK and IKK pathways in Huh7 cells in response to LPS. Cells were treated with LPS and collected at the indicated time points. Akt and JNK phosphorylation and IκBα degradation were analyzed by means of immunoblotting. The results are represented by three independent experiments. (**E**) The quantification of relative protein levels for p-Akt/Akt, p-JNK/JNK and IκBα in each condition. Error bars show standard deviation. (**F**) Activation of STAT3 in Huh7 cells with LPS treatment. Cells were treated with LPS for the indicated times. Cell lysates were analyzed by immunoblotting for STAT3 phosphorylation. The results are represented by three independent experiments. (**G**) Effect of LPS on HCC cell growth in vivo. SCID mice were inoculated with 2 × 10^6^ Huh7 cells s.c. in the right flank. Phosphate buffered saline (PBS) or LPS (2.5 mg/kg) was administrated i.p. once weekly, and treatment was started on day 1, when tumors were measurable (*n* = 6 for each group). Caliper measurements were performed to estimate the tumor volume (mean ± SD mm^3^). (** *p* < 0.01).

**Figure 3 ijms-23-08419-f003:**
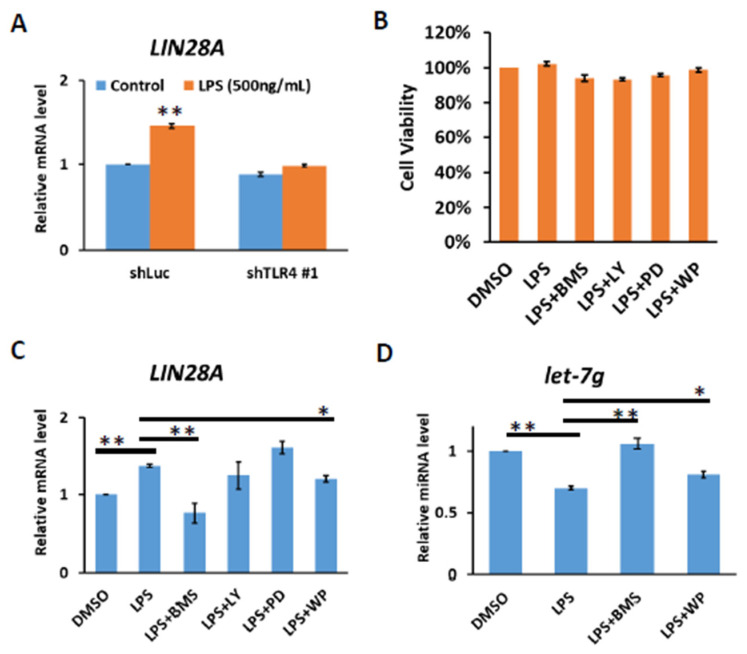
TLR4 activation induces IKK-dependent *LIN28A* expression. (**A**) Induction of LIN28A mRNA of Huh7 cells with or without TLR4 silencing in response to LPS. The cells were treated with LPS (500 ng/mL) for 4 h and RNA was extracted to be measured by qPCR. (**B**) Cell viability of Huh7 cells. Huh7 incubated with LPS (10 ng/mL) in the presence of indicated inhibitors was placed on 96-well plates for 4 h. Cell viability was measured with MTT assays. (**C**) *LIN28A* mRNA and (**D**) *let-7g* miRNA expression regulated by LPS depends on IKK and STAT3 pathways. Huh7 cells were treated with LPS (500 ng/mL) in the presence of indicated inhibitors, including IKK inhibitor (BMS345541 (20 μM)), PI3K inhibitor (LY294002 (20 μM)), MEK inhibitor (PD98059 (50 μM)) and STAT3 inhibitor (WP1066 (10 μM)), for 4 h. RNA was extracted, and the expression levels of the indicated mRNAs and miRNA were measured by qPCR. The relative mRNA and miRNA level in Huh7 cells without LPS treatment was set as 1.0. Results are mean ± SD for two separate experiments. (* *p* < 0.05, ** *p* < 0.01).

**Figure 4 ijms-23-08419-f004:**
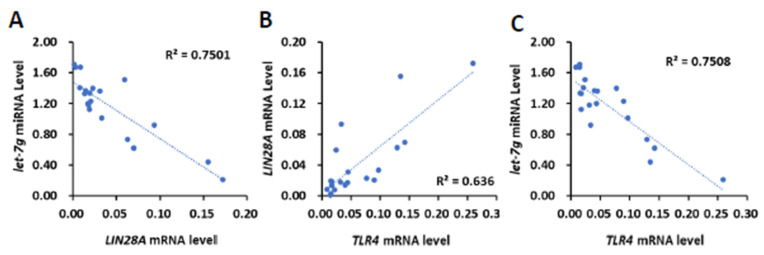
Positive correlation of *TLR4* with *LIN28A* mRNA and inverse correlation of *TLR4* and *LIN28A* mRNA with *let-7g* miRNA in clinical HCC samples. (**A**) Correlation between the expression of *LIN28A* mRNA with *let-7g* miRNA in HCC patients, and correlation between the expression of *TLR4* mRNA and (**B**) *LIN28A* mRNA and (**C**) *let-7g* miRNA. Pearson’s correlation analyses were performed. Data are presented as each value (R^2^, correlation coefficient).

**Figure 5 ijms-23-08419-f005:**
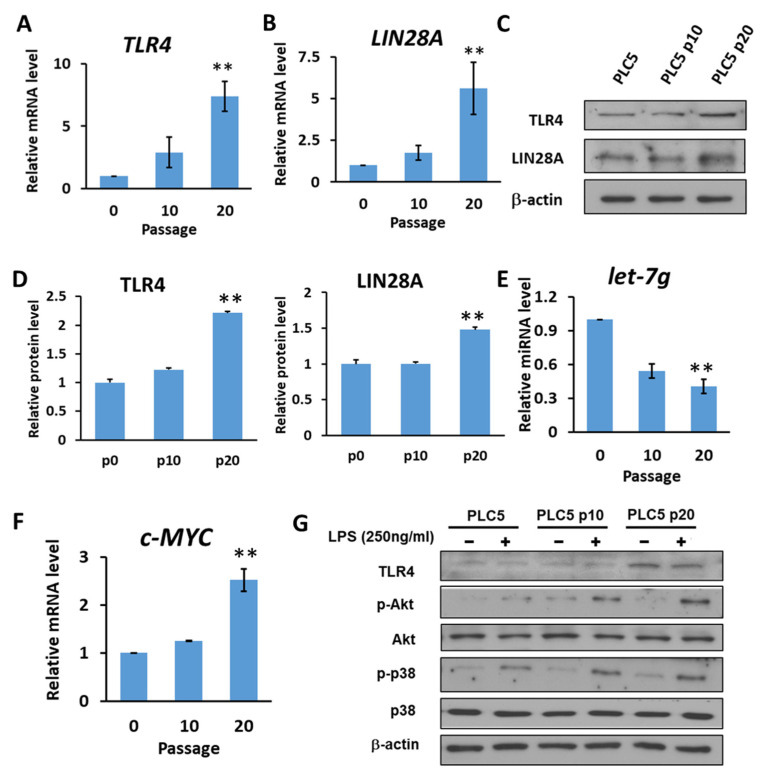
HCC with long-term LPS exposure leads to the upregulation of TLR4 and LIN28A, and the downregulation of *let-7g* miRNA. (**A**) *TLR4* and (**B**) *LIN28A* mRNA levels in PLC5 cells with or without long-term LPS exposure for indicated passages. (**C**) Protein level of TLR4 and LIN28A in PLC5 cells with LPS exposure for indicated passages. The expression levels of indicated proteins were detected with immunoblotting. The results are represented by three independent experiments. (**D**) The quantification of TLR4 and LIN28A protein levels compared to the α-actin control in each condition. Error bars show standard deviation. (**E**) The expression of *let-7g* miRNA level in PLC5 cells with or without LPS exposure for indicated passages. (**F**) *c-Myc* mRNA level in PLC5 cells with or without LPS exposure for indicated passages. The mRNA and miRNA levels were detected with qPCR. The relative mRNA or miRNA level in PLC5 cells without LPS treatment was set as 1.0. Results are mean ± SD for three separate experiments. (**G**) Activation of Akt and p38 pathways in PLC5 cells with or without long-term LPS exposure. Cells were treated without or with LPS (250 ng/mL) for 30 min. Akt and p38 phosphorylation was analyzed by means of immunoblotting. The results are represented by two independent experiments. (** *p* < 0.01).

**Figure 6 ijms-23-08419-f006:**
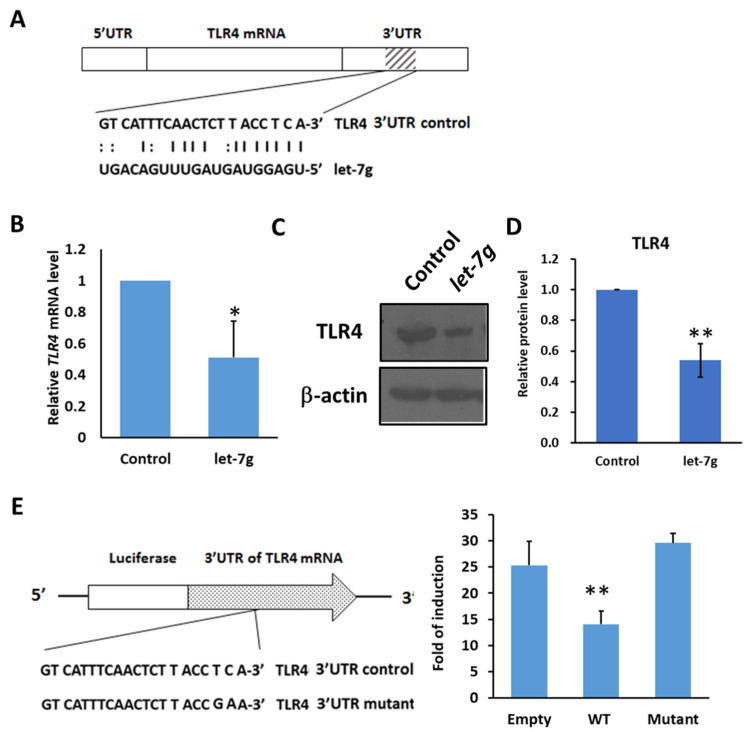
*TLR4* mRNA is targeted by *let-7g* miRNA. (**A**) Schematic of the seed region match between hsa-let-7g and the putative TLR4 3′UTR. MicroRNA.org was used to analyze and predict the possible seed region of has-let-7g on the 3′UTR of human *TLR4* mRNA. (**B**) Huh7 cells were transfected with control or hsa-let-7g precursor-expressed vector and incubated for 1 to 3 days as indicated. The levels of endogenous *TLR4* mRNA were quantified with qPCR. (**C**) Huh7 cells were transfected with control or hsa-let-7g miRNA precursor-expressed vector, incubated for 2 days, and harvested for the Western blotting of TLR4 protein levels. The results are represented by two independent experiments. (**D**) The quantification of relative protein levels for TLR4 in each condition. Error bars show standard deviation. (**E**) The post-transcriptional regulation of *TLR4* mRNA by *let7-g* miRNA with luciferase reporter assay. The positions of three seed match sites for *let-7g* and *TLR4* 3′UTR in the luciferase reporter construct pmirGLO-*TLR4*-3′UTR are indicated (**left**). Huh7 cells in 96-well plates were co-transfected with pre-let-7g miRNA-expressed vector and the indicated reporter constructs, including empty, wild-type *TLR4* 3′UTR, and mutated *TLR4* 3′UTR, respectively, in the downstream of Firefly luciferase gene. Firefly and Renilla luciferase activities were measured at 24 h post-transfection. The data represent the means from three independent experiments. (* *p* < 0.05, ** *p* < 0.01).

**Figure 7 ijms-23-08419-f007:**
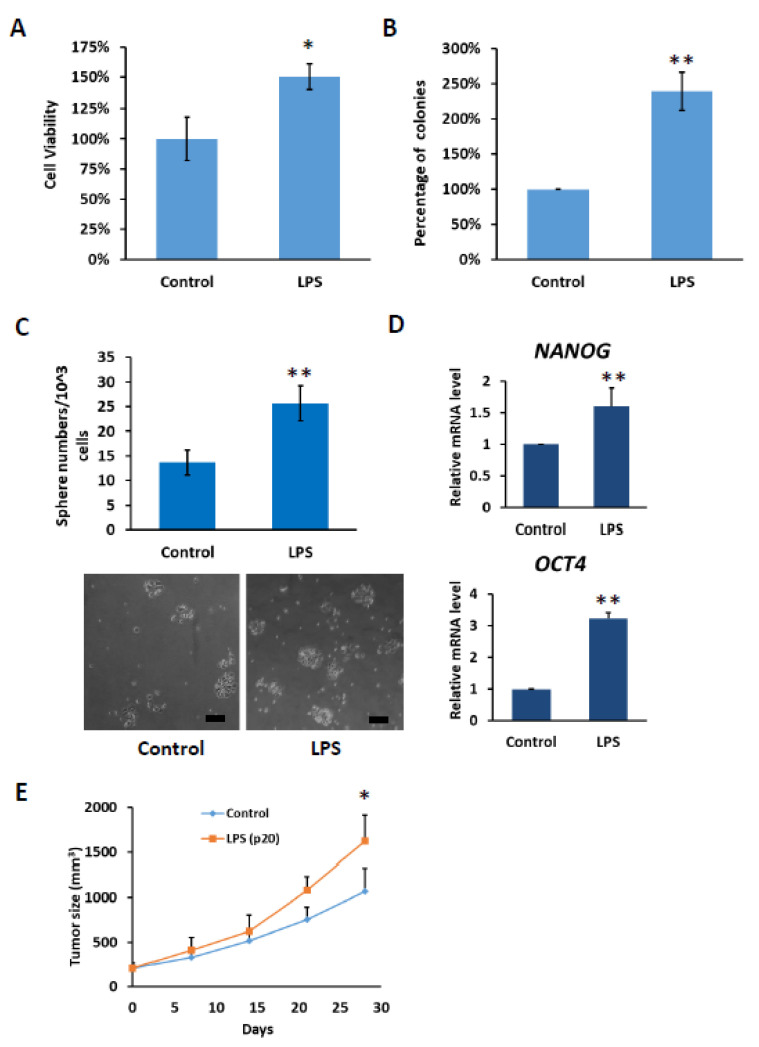
Cell growth and stemness properties were increased in PLC5 cells with long-term LPS exposure. (**A**) Cell viability and (**B**) clonogenic ability of PLC5 cells under long-term LPS exposure. PLC5 cells with or without LPS exposure for 20 passages were placed on 96-well plates for 4 days or on 6-well plates for 14 days. Cell viability was measured with MTT assays. Clonogenic ability was determined by colony formation, which was visualized by crystal violet. The relative viability and clonogenic ability of PLC5 cells without LPS treatment were set as 100 percent. Results are mean ± SD for three separate experiments. (**C**) Non-adherent sphere formation of PLC5 cells under long-term LPS exposure. PLC5 cells with or without LPS exposure for 20 passages were plated on 96-well plates and incubated for 4 days. Bright-field images are shown. Scale bar = 100 μm. Number of large spheres generated from 1 × 10^3^ PLC5 cells with LPS exposure for indicated passages was quantified. Results are mean ± SD for three separate experiments. (**D**) The levels of *OCT4* and *NANOG* mRNA in spheres from PLC5 cells with or without LPS exposure. The mRNA levels were detected with qPCR. (**E**) Effect of long-term LPS exposure on HCC cell growth in vivo. SCID mice were inoculated with 5 × 10^6^ PLC5 cells with or without LPS exposure for 20 passages s.c. in the right flank (*n* = 6 for each group). Caliper measurements were performed to estimate the tumor volume (mean ± SD mm^3^). (* *p* < 0.05, ** *p* < 0.01).

**Figure 8 ijms-23-08419-f008:**
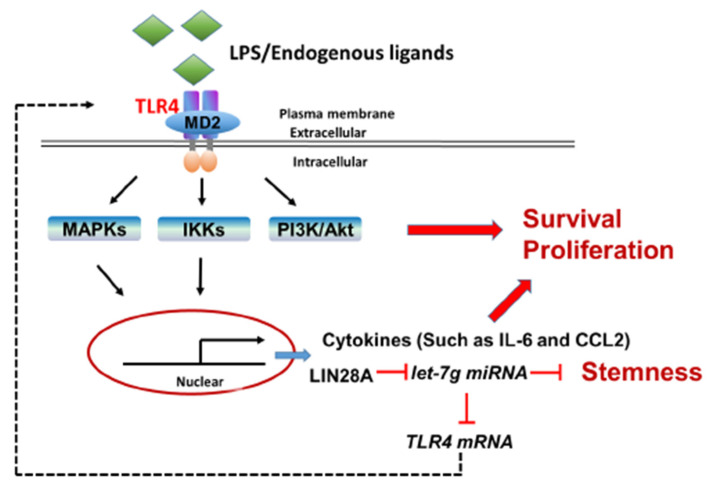
Proposed model for the positive feedback loop of TLR4-mediated survival signaling in liver cancer.

## Data Availability

The data presented in this study are available upon request from the corresponding authors.

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
