# Peer review of "Persistent TLR4 Activation Promotes Hepatocellular Carcinoma Growth through Positive Feedback Regulation by LIN28A/Let-7g miRNA"

_ijms, 2022, doi:10.3390/ijms23158419_

Round 1
Reviewer 1 Report
The authors have positively met our previous criticism
Author Response
Thanks for reviewer's previous criticism.
Reviewer 2 Report
Comments to the authors:
The research article by Chen et al., aims to explore the regulatory mechanisms that control TLR4 activation and their potential implication during liver carcinogenesis. The manuscript presents multiple grammatical mistakes, which increases the difficulty of understanding the main message of this manuscript. More importantly, I consider that additional experiments are required in order to validate the model proposed by the authors, as a number of the regulatory links between these factors are based on correlations.
Major comments:
- The manuscript includes many grammatical mistakes. As pointed out by the previous reviewers, I also consider that the article should be revised by a skilled writer. Format mistakes such as the use of abbreviations and definitions also need to be corrected. I would have included my suggestions, but the document does not include line numbers, which makes communication with the authors difficulty.
- In figure 3A the authors state that LPS induces an increase of LIN28A. This does not seem to be the case, as LPS treatment did not have an effect in combination with shLuc or shTLR4. These results contradict the ones presented in figure 3B. Moreover, the authors did not include cell viability data in order to show that the effect of the compounds used in figure 3B is not related to cell toxicity. These points need to be addressed by the authors.
- According to the model put forward by the authors, LIN28A induces downregulation of let-7g. This has not been demonstrated by the authors, as only correlations are presented in support of this regulatory link. It would be interesting to evaluate if overexpression of LIN28A induces downregulation of let-7g.
- The results presented in the article do not support the following conclusion: “Importantly, we demonstrate that elevated TLR4 expression caused by chronic inflammation in HCC promotes tumorigenesis through activation of STAT3 signal and downregulation of let-7 miRNA”. STAT3 inhibition was not performed in any of the presented experiments in order to support a direct link to the phenotypes observed.
Minor comments:
- The reviewer considers that is quite a challenging task to obtain solid conclusions based on some of the western blots presented in this article. A clear example of this is the figure 6C, in which TLR4 bands are barely observed. Moreover, original western blots of the conditions without LPS are missing for the supplementary data of figure 5C.
Reviewer 3 Report
The authors investigated the effect of LPS on growth and proliferation of HCC cells. They demonstrated that the LPS/TLR4 axis activates proliferative signaling cascades and inflammatory cytokine expression. Moreover, it induced the expression of LIN28A, which downregulated LET7G expression. Because LET7G suppresses TLR4 expression, LPS treatment enhanced the expression of TLR4 in HCC cells. The authors showed correlation in these genes' expression in clinical HCC samples. These findings are interesting. However, the data are preliminary, and further experiments are necessary to support the conclusion of this manuscript.
1. Introduction: Cigarette smoking is not a main risk factor of HCC. Rather, metabolic syndrome, high calorie intake, diabetes, and obesity are more important.
2. Supplementary Figure S2: The following sentence should be revised (HepG2 and Hep3B are appeared twice while Sk-Hep1 is not):
"HepG2 and Hep3B, Huh7 cells expressed high level of TLR4, and PLC5, HepG2 and Hep3B cells are with relatively low TLR4 expression (Supplementary Figure S2)."
The authors should separately explain mRNA and protein expression levels. In particular, Hep3B shows inconsistent expression levels between mRNA and protein. This point should be clearly mentioned.
3. The knockdown efficiency of shTLR4s should be shown.
4. Figure 2CDEF: All or some of these effects of LPS were tested in PLC5 and/or TLR4KD HuH7 cells to confirm that they are elicited by the LPS/TLR4 pathway.
5. Figure 3B: It should be investigated whether LET7B expression was induced in the presence of BMS.
6. Figure 5: It should be investigated whether MAPK, PI3K/AKT, JNK, NFkB and inflammatory cytokines are altered by the long-period treatment with LPS.
7. Figure 5: It is necessary to investigate whether these alterations are attributable to TLR4/LET7G by knocking down TLR4 or transfection of LET7G in PLC5 p20 cells.
8. Figure 5E: Although c-MYC could be a factor of cancer stemness, it also has other functions including proliferation. Because there are more important and reliable factors, such as OCT4, NANOG, SOX2, KLF4, CD13, CD44, CD90, CD133 etc., the expression of them should be investigated in 2D and spheroid culture.
9. Figure 6C: Band image is not clear to see the difference in TLR4 expression.
10. Figure 7: As I indicated in #6, the involvement of TLR4 and LET7G should be investigated.
11. The authors investigated the effects of LPS on cell growth; however, it is better to investigate whether and how LPS affects tumor immunity.
Round 2
Reviewer 3 Report
The authors have adequately addressed my concerns and improved the manuscript.
I have no further comments.
Author Response
Thanks for reviewer's comments
This manuscript is a resubmission of an earlier submission. The following is a list of the peer review reports and author responses from that submission.
Round 1
Reviewer 1 Report
The authors investigate the mechanism linking chronic TLR4-mediated inflammation and the development of hepatocellular carcinoma. Quite a lot is already known about the progression of HCC and the involvement of inflammation involving TLR4, as evidenced by the existence of numerous published articles. The role of TLR4 in stimulating HCC cell growth is well known, so this information in the present article is of little novelty. The authors should refer to the already known TLR4-activated pathways in HCC. The hypothesis and objectives are not well defined. The most novel part of the paper is the relationship between TLR4 and stemness characteristics of HCC cells.
1) The authors mention in the introduction that the role of TLR4 expression and inflammation in HCC is not well understood. But there are numerous published articles, such as Dapito, Cancer Cell 2012, 21: 504 or Zheng, Theranostics 2020, 10: 9923, among others.
2) The hypothesis and objectives are not well defined. Instead, at the end of the introduction, the authors merely tell the process they have followed, but it is not a real hypothesis.
3) The results shown in Figures supp. 2 and 3 and Fig. 2 are already well known, even in in vivo experiments. They add little to the work.
4) The connection between TLR4 activation and LIN28 expression has not been studied in detail and is one of the most important parts of the work. Pathways such as MAPK, IKK, PI3K are discussed. Studies with pathway inhibitors or shRNA are needed to elucidate which one is involved. In addition, the NF-kB or FGF21 pathways, well known from the TLR4 pathway, have not been considered.
5) As the authors already did in the experiments in Figure 2, to demonstrate the relationship between TLR4 and tumor growth, the same should be done to demonstrate the relationship between TLR4 and LIN28. For example, by TLR4 knockout.
6) Only the relationship between TLR4 and the cytokines IL-6 and CCL2 has been studied, but the most known cytokines following TLR4 activation are TNFa and IL-1b.
Author Response
We thank the reviewer for finding our work interesting and for the constructive criticisms that have greatly improved our earlier conclusions.

Reviewer 2 Report
In this paper is confirmed the overexpression of mRNA expression of TLR4 and proinflammatory cytokines, including IL6 17 and CCL2 in HCC. A positive correlation between TLR4 and LIN28A and a negative correlation between let-7g miRNA with TLR4 or LIN28A was demonstrated. In PLC5 cells cultured in low-dose LPS-containing medium to mimic chronic inflammation for persistent TLR4 activation, high levels of TLR4 mRNA and protein were found The increase of LIN28A led to a decrease of let-7g miRNA. 3’UTR of TLR4 mRNA was shown to be the target of let-7g miRNA. Long-term LPS exposure increased the PLC5 cells proliferation, cytokine expression and stemness properties. In conclusion this study shows n HCC a positive feedback regulation for chronic TLR4 activation to modulate TLR4 expression level through LIN28A/let-7g pathway . This paper contains some drawbacks. Activation of Akt, JNK and IKK pathways in Huh7 cells in response to LPS is not evident. The quantitative evaluation of western blots and the statistical analysis of the data in Fig, 2D are needed. Does exists a correlation between the expression of these genes and human HCCs? The genotype and the phenotype of human HCCs are extremely variable. Therefore, a number elevated of samples should be analyzed to determine the clinicopathological features of HCC patients. Non-adherent sphere formation of PLC5 cells in Fig. 6 are hardly visible in the PDF copy of the paper.Author Response
We thank the reviewer for appreciating the potential importance of our findings and for pointing out those areas that required further clarifications and more experimentation.
